# MRI Bone Abnormality of the Knee following Ultrasound Therapy: Case Report and Short Review

**DOI:** 10.3390/ijerph192114202

**Published:** 2022-10-30

**Authors:** Ismaël Moussadikine, Mỹ-Vân Nguyễn, Christophe Nich, Pierre-Paul Arrigoni, Yonis Quinette, Vincent Crenn

**Affiliations:** 1Orthopedic and Traumatology Unit, Nantes University Hospital, 1 Place Alexis Ricordeau, 44000 Nantes, France; 2INSERM, UMRS 1229, Regenerative Medicine and Skeleton (RMeS), Nantes Université, ONIRIS, 44042 Nantes, France; 3Radiology Department, Nantes University Hospital, 1 Place Alexis Ricordeau, 44000 Nantes, France; 4CRCI2NA (Centre de Recherche en Cancérologie et Immunologie Nantes-Angers), INSERM UMR 1307, CNRS UMR 6075-Team 9 CHILD (Chromatin and Transcriptional Deregulation in Pediatric Bone Sarcoma), Nantes Université, 1 rue Gaston Veil, 44035 Nantes, France

**Keywords:** ultrasound therapy, MRI, bone abnormality

## Abstract

Ultrasound (US) therapy in sports and medical pathologies is widely used by many physiotherapists and sports medicine clinicians; however, data regarding their potential side effects remain rare. We report a case of a 21-year-old woman with iliotibial band (ITB) syndrome treated with a physiotherapy session combined with US therapy. She had twenty 7 min US sessions on the knee, for 3 months (US at 1 Mhz with an intensity between 1 and 2 W/cm^2^). Due to persistence of the ITB syndrome’s symptomatology after the 3 months of physiotherapy sessions, an MRI (magnetic resonance imaging) was carried out and revealed osteonecrosis-like bone abnormalities on the external femoral condyle, the external tibial plateau, and the proximal fibula. In view of these lesions, the ultrasonic therapy was stopped, and a repeat MRI demonstrated the progressive disappearance of these imaging abnormalities one year after the last US (ultrasound) treatment. In light of this case, we propose here a short review of reported osseous “osteonecrosis” abnormalities associated with US therapies.

## 1. Introduction

Ultrasound therapies are used increasingly in sports medicine, especially in various ligament and muscle injuries. The therapeutic applications of US predate its use in imaging and can be used at “low power” or “high power”. Depending on the intensity and duration of exposure, a vast spectrum of biological changes is achieved. At low intensities (∼100 mW/cm^2^), any effect observed is likely to be reversible and/or beneficial for tissue healing, as in sports medicine. At the other end of the spectrum, very high intensities (∼1000 W/cm^2^) are capable of producing instantaneous tissue necrosis, and its use is actively explored in oncology. As early as the 1950s, “low power” use was described in physiotherapy, fracture repair, sonophoresis, sonoporation, and gene therapy, while high-powered applications include high-intensity focused US (HIFU) and lithotripsy [1,2,3].

The widespread use of US therapy, with no known dose–response relationship, is an issue. US therapy users can only guess what dose might be useful for a patient and how to modify it. The level of clinical benefit to the patient from US physiotherapy treatments therefore remains uncertain, and more clinical research is needed to justify the dosages currently used in treatment [4,5].

There are few articles related to the side effects, more precisely bone abnormalities or “osteonecrotic-like” lesions of US used for therapeutic purposes. Here, we report an example of a lesion that did not lead to clinical consequences but instead to the appearance of a noticeable unusual abnormality on MRI. Our case report and short review of the literature thus focus on these radiological abnormalities following US therapy.

## 2. Case Report

The patient was a 21-year-old Caucasian woman with no particular family, medical, or psychosocial history. She had been complaining for two months of pain on exertion in the context of tendinopathy of the ITB syndrome type secondary to practicing indoor exercise (musculation, cross-fit,…). On clinical examination, she presented no pain on palpation over the knee except for the lateral knee ITB area, no limitation of mobility, and no patellar shock or instability of the knee. The patient had never received corticosteroids either systemically or locally. An MRI was performed one month before the US therapy, which identified a bone reaction in regard to the ITB limited to the lateral facet of the lateral femoral condyle (Figure 1).

In this context, the patient started a physical therapy protocol including stretching, eccentric work, and 20 US sessions, on the knee’s external face, for 3 months. These were 7 min sessions with US at 1 Mhz with an intensity between 1 and 2 W/cm^2^. Given the persistence of her initial symptoms, without any modification to the physical examination, a further MRI was carried out one month after the last US therapy (Figure 2). This MRI highlighted multiple and extended signal abnormalities under the epiphyseal cortex with the peripheral border in hyper-intensity in DP FS sequences, and hypo-intensity in T1 sequences with respect to the fatty signal within the anomalies and a significant extension of the lesions, both concerning the lateral femoral condyle, but also the lateral face of the tibial plateau and within the proximal fibula.

Three months after the first MRI, the patient presented a decrease in her symptoms, with no pain, no limitation in joint mobility, and no reduction in muscle strength with a stable knee in the frontal and sagittal plane. The X-ray was normal (Figure 3).

A further MRI performed 5 months after the previous one (6 months after US therapy ended) showed an apparent reduction in the bone abnormality lesions of the lateral femoral condyle, the lateral tibia plateau, and the proximal fibula (Figure 4).

Due to the chronology, these lesions were utterly compatible with the bone remodeling connected with US. In fact, these were lesions occurring in a non-weight-bearing area next to the treated area. No other support in rehabilitation was carried out after the ultrasounds and no restriction after the end of the ultrasound sessions. At 15 months after the last US therapy, the patient no longer presented any pain or joint limitation with her knee. She was able to return to sports activities. Moreover, we observed modifications on MRI imaging with an almost complete disappearance of the “osteonecrosis-like” lesion (Figure 5). However, the lesion initially found on the MRI before US therapy persisted in the same location with each new imaging performed.

## 3. Discussion

This specific case leads to some questions about US therapy. First, we will recall iliotibial tendinopathy syndrome and discuss the osteonecrosis knee. Then, we will focus on the effects of US, and its uses in therapeutic cases, and discuss bone abnormalities and “osteonecrosis-like” lesions reported in the literature associated with US therapy. Finally, we will draw conclusions regarding the role of this tool in the treatment of musculoskeletal injuries, especially in sports medicine.

### 3.1. Knee Lesions Encountered

#### 3.1.1. Iliotibial Band Syndrome

The ITB syndrome presented by our patient is a common, painful condition caused by inflammation of the distal part of the ITB that occurs on the lateral side of the knee. The first insertion of the iliotibial band is in the distal femur at the upper edge of the lateral epicondyle. The second attachment of the iliotibial band is the insertion into Gerdy’s tubercle of the tibia and serves as a ligament in its structure and function. The iliotibial band has many other distal attachments (the biceps femoris, vastus lateralis, lateral patellar retinaculum, patella, and patellar tendon). The site of injury is often associated with insertion at the lateral epicondyle but interrelated with the forces created by the various attachments above and below the lateral epicondyle.

Repetitive flexion and extension movements during sports exercise, along with excessive friction of the distal ITB, lead to its irritation and inflammation, as it slides over the lateral femoral epicondyle causing friction, irritation, and lateral femoral pain [6,7]. It is usually seen in people with intense physical activity, such as athletes. It seems to be the most common running injury of the lateral knee region, with an incidence of 1.6–12% [8,9]. The kinetics and kinematics of the hip, knee, and ankle also seem to play an important role in ITB syndrome, and neuromuscular coordination emerged as a likely reason for the kinematic defects. Thus, several intrinsic and extrinsic contributing factors to ITBS have been described. Reduced hip muscle performance and abnormal hip and knee mechanics during functional tasks may be major contributors to ITBS.

The diagnosis of ITB is mainly based on history and physical examination. Without any other pathology, the standard knee X-rays will appear normal. If the diagnosis is still unclear after the history and physical examination, an MRI of the knee can confirm the diagnosis if it shows hyperintensities in the lateral epicondyle with a distally thickened ITB. Ultrasound is an inexpensive, low-risk modality that may also show the abnormal distal thickening of the ITB [10,11]. Some publications have reported MRI findings of signal changes with bone edema and subchondral bone erosion, with or without associated soft tissue damage immediately below the ITB, without signs of inflammation or thickening of the band itself. However, these signal intensity abnormalities are sometimes poorly visualized in a compartment bounded laterally by the ITB [12,13,14].

Regarding therapeutic management of this syndrome, many strategies are suggested apart from the US therapy used in our case. Physiotherapy (stretching of the ITB strengthening of the adductors, improvement of muscle coordination) in association with limited sports activities, and oral nonsteroidal anti-inflammatory drugs are typically described in first-line treatment. In cases where this is ineffective, some authors might recommend anti-inflammatory injections, or even surgical treatment in the management of this syndrome [15,16,17,18]. Finally, US therapy has been described by some authors in physiotherapy sessions in association with deep transverse friction using various protocols [19].

#### 3.1.2. Osteonecrosis Lesions of the Knee

As the lesions occurring during US treatment in our case report had similarities with authentic osteonecrosis lesions, it seems relevant to recall the precise characteristics of the latter. Osteonecrosis of the knee, mainly caused by alterations in the bone blood supply, was first described by Ahlback et al. in the 1960s [20]. After the hip, the knee is the second most commonly affected location [21].

Various entities include acute traumatic osteochondral lesions, subchondral insufficiency fractures, so-called spontaneous osteonecrosis of the knee, avascular necrosis, osteochondritis dissecans, and osteochondral abnormalities localized in osteoarthritis [22]. Osteonecrosis of the knee itself is a poorly understood and disabling disease. Originally described as a single disorder, it encompasses three distinct conditions: spontaneous osteonecrosis of the knee (SPONK), secondary osteonecrosis of the knee (due to systemic diseases or treatment side effects, etc.), and post-arthroscopic osteonecrosis of the knee [23].

In the knee, its typical presentation shows bone marrow lesions; MRI alteration classically describes a modification in cancellous bone signal intensity, with a high signal on fluid-sensitive sequences (T2), with or without a low T1 signal [24]. Authentic knee osteonecrosis is a progressive disease that often leads to subchondral collapse and disabling arthritis. Nonetheless, in the early stages of osteonecrosis, some authors described the potential evidence of reversal on the MRI, using conservative treatment (such as bisphosphonates, prostaglandin agents, enoxaparin, statins, hyperbaric oxygen, extracorporeal shockwave therapy, and pulsed electromagnetic field therapy), advocating for a potentially reversible nature [25,26]. This normalization of osteonecrosis lesions might be the same as that described in our clinical case when the US therapy stopped.

### 3.2. The Effects of Ultrasound Therapy

#### 3.2.1. US Vibration Principles

Ultrasonic vibrations have two effects: first, a thermal effect resulting from the molecular friction caused by the vibrations, leading to elevation of the peripheral nociceptor activation threshold and a decrease in neuromuscular spindle activity, which might promote the relief of bone, muscle, and joint pain. It also has a mechanical effect, due to vibrations that cause pressure variations, leading to the release of gas in the form of microscopic bubbles. This mechanical effect produces micro-cuts that cause, on the one hand, changes in cell permeability, favoring exchanges, and, on the other hand, a dilaceration of the fibers in the connective tissue, known as the fibrinolytic effect, used in the treatment of adhesions and scars [27,28]. These two actions of US devices are used to explain the analgesic actions of US on pain points (muscular, articular, neuralgic, epidermal), but also the fibrinolytic or defibrosing action, associated with relaxation, and anti-inflammatory action (by improving circulation) [27,28].

Ultrasound devices offer two frequencies: 1 MHz or 3 MHz [29,30]. The 1 MHz frequency is used to treat deep areas (about 5 cm deep) with US therapy, whereas the 3 MHz frequency is used to treat superficial areas (about 1.5 cm) [31,32]. The physical origin of hyperthermia is the absorption of US into the tissues. These are all the more important as the frequency is high [33]. In addition, as some studies have shown, diathermy uses specific forms of energy, such as microwave diathermy, to raise the temperature of the deepest soft tissues [34,35,36].

#### 3.2.2. Therapeutic Use of US

Several studies present the use of US therapy. It was first explored by Wood and Loomis as early as the late 1920s [37]. They highlighted that the most important parameter for US, besides frequency, was intensity, with various effects.

Diagnostic imaging applies US at intensities ranging from 0.05 to 0.5 W/cm^2^, while high intensities are used notably in surgery, ranging up to 10,000 W/cm^2^ [38,39,40]. Therapeutic US with high intensities primarily uses its thermal action, while the effectiveness of low-intensity treatments is predominated by non-thermal effects, including acoustic cavitation [41].

Low-intensity US generally applies the frequencies of 1–3 MHz, with intensity ranging from 0.02–1 W/cm² and has a variety of therapeutic applications. In bone healing, some authors report reduced fracture healing time, particularly in delayed unions and non-unions [42,43]. The US effect on fracture healing might increase the amount and strength of bone callus, by means of several biological and molecular mechanisms [44,45,46]. Nonetheless, objective quantification of these effects is difficult due to the high heterogeneity in the parameters and protocols. Some recent articles have shown a lack of clear clinical benefit of low-intensity US, thus questioning its effectiveness [47,48].

US therapy is also used by various teams in soft tissue regeneration, with potential effects in promoting tendon and ligament healing, and cartilage recovery. It can also be used in inflammation inhibition, neuromodulation, and dental treatment according to various authors [49,50]. Overall, the levels of evidence are very low, and these areas remain less studied than bone healing. As with bone healing, more randomized human clinical trials with unified controlled parameters appear necessary to receive more systematic data to support therapy both alone and in combination with other stimulation methods.

The literature on knee arthritis is no more robust, with very little evidence supporting therapeutic US; however, the delivery method of US (pulsed vs. continuous) is controversial. No conclusive recommendations can be made for optimal settings or session duration [51]. As we stated previously, there is no clear benefit for US therapy in musculoskeletal disorders, with very low evidence. Despite this evident lack of clear recommendations and objective effectiveness, US is widely used by physiotherapists and in sports medicine around the world [52,53,54,55,56].

#### 3.2.3. US Bone Level Consequences

(1)Hypothesis of bone side effects

In view of the potential effects explained above, it is therefore quite conceivable that the effects of US therapy may have consequences on the different tissues, and in our case, consequences at the bone level. Unfortunately, very few studies have described such negative effects [57,58].

A common conclusion of surveys conducted since 1974 on the clinical use of US therapy devices was that these devices were generally unable to deliver prescribed doses to patients with any reasonable degree of accuracy. This occurred because the indicated output of the device was often unrelated to the actual acoustic output. Excessive exposures either result in unnecessary risk or fail to achieve clinical benefit. As the intensity levels used in US therapy are within the range where adverse biological effects have been observed in animal studies, treatment doses must be accurately indicated and delivered [57,58].

An essential feature of the regulations is that the maximum temporal average of sufficient US intensity should not exceed 3 W/cm^2^. This value was chosen for several reasons: (i) higher intensities do not appear necessary for clinical effectiveness on healing tissues; (ii) 3 W/cm^2^ is commonly found as the maximum nominal intensity available for most appliances, and European manufacturers have accepted this intensity value for many years as the maximum necessary for therapy; (iii) higher intensities can be painful or damaging [59].

In addition, although US therapy is frequently used as an electrophysical agent in sports medicine, systematic reviews and meta-analyses have concluded that there is insufficient evidence to support the beneficial effect of US at the dosage currently used in clinical practice [56,60]. A question therefore arises: can these US used for therapeutic purposes have side effects?

Too few details are still provided in most studies to identify a relationship between dosage and response to treatment [56,61,62], with high protocol disparities between each study. With no known dose–response relationship, US therapy users can only guess what dose might be sufficient for a patient and how to modify it. Much more clinical research is needed to justify the dosages currently used in treatment.
(2)“Osteonecrosis-like” lesions reported in the literature

Despite US intensities below the “maximum dose” described above, and four months’ duration, in our case the patient presented with “osteonecrosis-like” lesions, probably linked to those ultrasound sessions. This advocates for US having an authentic biological effect on bone tissue. These concerns have already been raised by others.

Lee-Ren Yeh’s team revealed subcortical lesions of the knee, the shoulder, and the wrist in MRI with eight patients who also had persistent pain or symptoms [63]. The imaging characteristics of the lesions were similar to those of focal osteonecrosis. Follow-up MRI in three patients from 2 to 3 months after stopping US therapy revealed the resolution of the bone lesions, as in our case. This study demonstrated that US diathermy may result in bone damage. The abnormality itself was generally mild and transient, with apparently complete recovery after the end of therapy [63]. Pre-ultrasound diathermy procedure MRI image evaluations were not available in the study by Lee-Ren Yeh et al. and thus do not allow the authors to formalize the causal role of US therapy in the genesis of these lesions. Nonetheless, this study on eight cases aligns with our hypothesis of osteonecrosis-like lesions induced by US therapy.

In the same way, in their two patients treated with US, Seung Jae J. Kim et al. described diathermy-associated focal bone marrow abnormalities of the superolateral humeral head [64]. They concluded that the timing and transient nature of the findings in relation to the diathermy procedures suggest that US diathermy was the cause of these focal bone marrow lesions.

Preclinical models have also been studied on this topic. Smith et al. showed that focus on the US directed at a rabbit’s femur caused immediate and significant thermal damage to the bone, in the form of osteocyte necrosis. They therefore concluded that when the focused US energy is directed at or near the bone–muscle interfaces, care must be taken to avoid thermal damage to the bone, which can compromise its strength for long periods [65]. Histological and hematological studies conducted on the femurs of six dogs that had received different intensities of US, at 1.5 and 2.5 W/cm^2^, gave presumptive evidence of bone marrow damage and regeneration [58].

These various studies identify an US effect specific to bone tissue, with osteonecrotic-like bone consequences in human cases, as well as in preclinical models.

### 3.3. What Is the Role of US Treatment in Musculoskeletal Disorders?

Finally, as seen in our review, there are presumptions and preclinical evidence of certain effects of US therapy, but evidence of clinical benefits is lacking. Nonetheless, there are indeed biological effects of this treatment, as seen with bone osteonecrotic lesions in our case, and in the current literature.

After reviewing 293 articles published since 1950 to evaluate the evidence for the effect of the US on treating musculoskeletal disorders, Cam et al. concluded that using the US to treat musculoskeletal disorders is based on practical experience but lacks strong evidence from well-designed controlled studies. It remains to be seen whether US treatment can increase the effectiveness of exercise therapy in musculoskeletal disorders [66]. Furthermore, as pointed out by Dewhirst, there is a lack of studies on the type and extent of the damage occurring in the temperature range following US therapy [67]. In France, there are no formal guidelines concerning the use of US therapy in sports medicine, in myotendinous pathologies. Despite this, this therapy remains widely used by a significant number of physiotherapists.

Our case shows that bone osteonecrosis-like lesions can appear due to US therapy. Our MRI studies and chronology made it possible to observe the evolution of these lesions, with the gradual disappearance of the lesions at the end of the US treatment. This phenomenon allows us to strongly suspect the causality of US therapy in the appearance of bone osteonecrosis-like lesions.

## 4. Conclusions

In conclusion, this case suggests US therapy may be associated with localized “osteonecrosis-like” bone lesions, as seen on MRI. These lesions were not associated with any clinical symptoms, and they disappeared over time once the US treatment had stopped. Thus, despite wide use of US therapy, there is limited evidence of its effectiveness, and it may instead induce bone damage.

## Figures and Tables

**Figure 1 ijerph-19-14202-f001:**
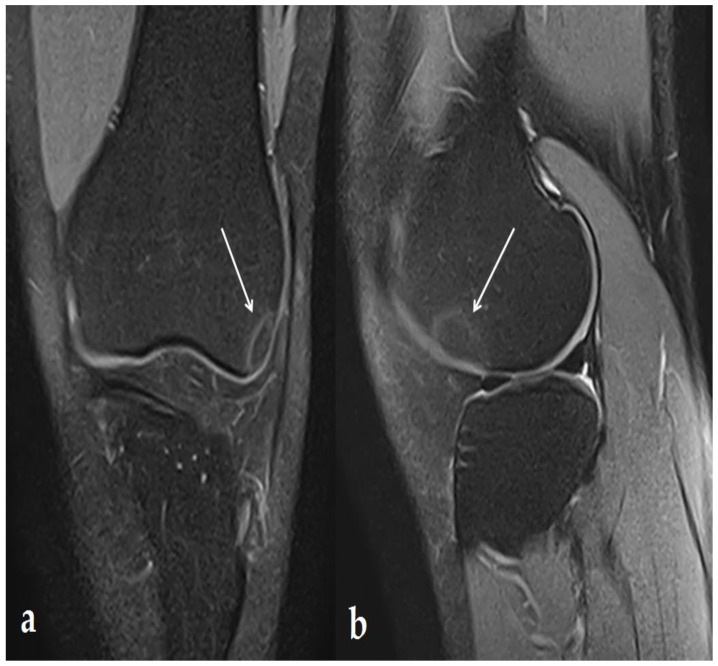
MRI: 1 month before the first US therapy. DP FS MRI image in frontal (**a**) and sagittal (**b**) section demonstrating a single (13 × 14 × 16 mm) lesion under the epiphyseal cortex of the lateral femoral condyle with peripheral border in hyper-intensity, highlighted with white arrows.

**Figure 2 ijerph-19-14202-f002:**
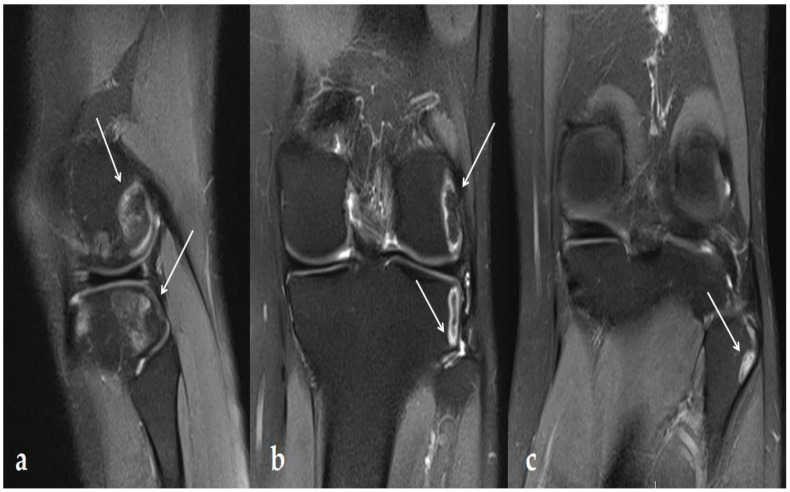
MRI: 1 month after the last US therapy. DP FS MRI image in sagittal (**a**) and frontal (**b,c**) sections demonstrating signal abnormalities under the epiphyseal cortex with hyper-intensity in the peripheral border: 21 × 20 × 12 mm lesion on the femur, 16 × 18 × 15 mm on the tibia, and 10 × 9 × 9 mm on the proximal fibula, highlighted with white arrows.

**Figure 3 ijerph-19-14202-f003:**
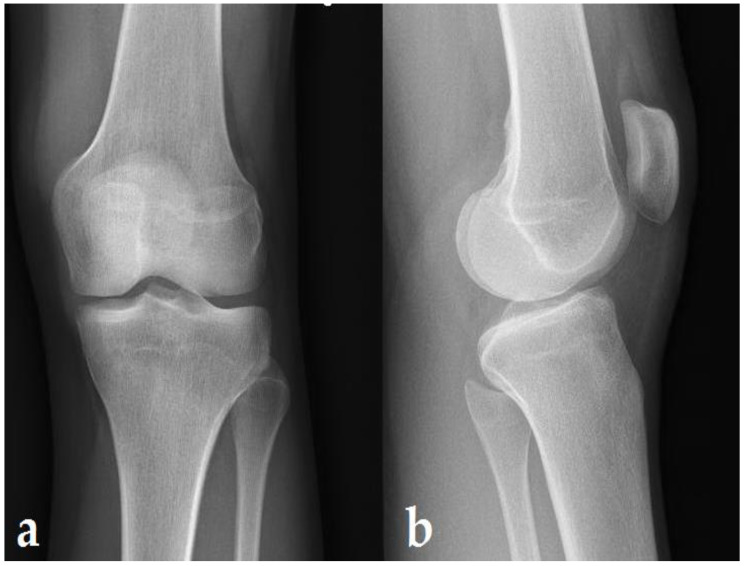
Standard X-rays: 4 months after the last US therapy. AP (**a**) and lateral view (**b**) showing no bone lesion.

**Figure 4 ijerph-19-14202-f004:**
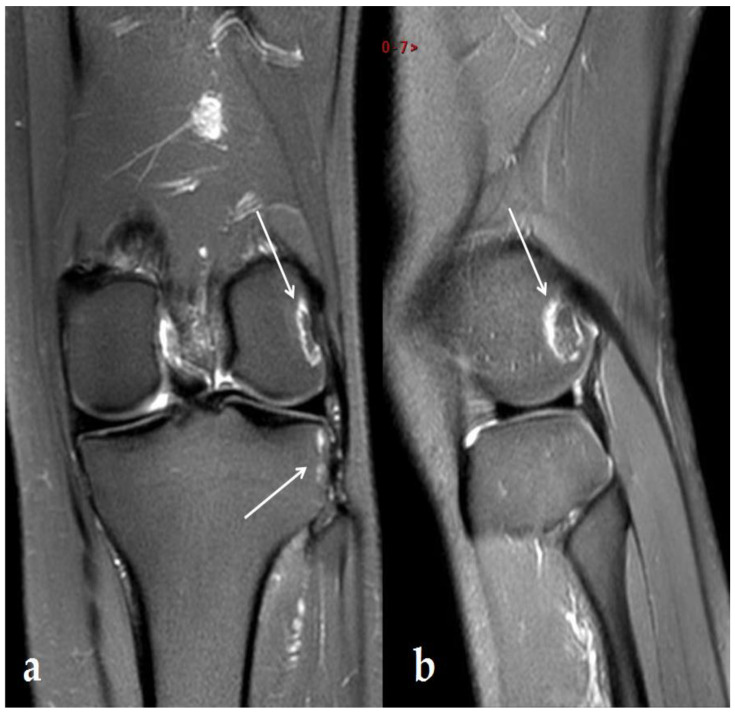
MRI: 6 months after the end of US therapy. DP FS MRI image in frontal (**a**) and sagittal (**b**) sections showing a reduction in lesions of the lateral femoral condyle, lateral tibial plateau, and head of the fibula with a sinuous cortical-to-cortical necrotic-like demarcation border in DP FS hypersignal 6 months after the US stopped. An 11 × 11 × 8 mm lesion remained on the femur and a 14 × 14 × 8 mm lesion on the tibia, but there was complete disappearance on the fibula. Arrows show lesion.

**Figure 5 ijerph-19-14202-f005:**
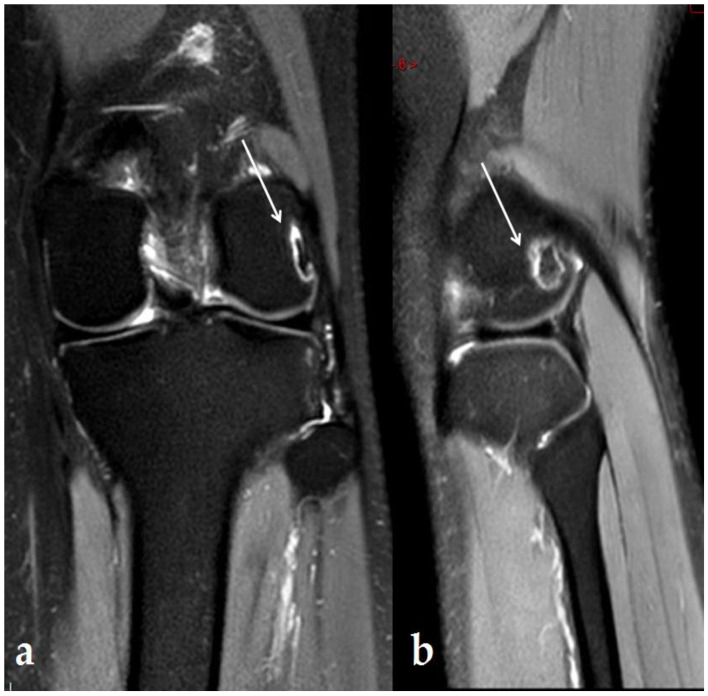
MRI: 11 months after the last US therapy. DP FS MRI image in frontal (**a**) and sagittal (**b**) sections showing remaining bone abnormality. At the last follow-up, only a single lesion persisted on the femur of 8 × 9 × 5 mm. arrows show lesion.

## Data Availability

Not applicable.

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
