# Peer review of "MRI Bone Abnormality of the Knee following Ultrasound Therapy: Case Report and Short Review"

_ijerph, 2022, doi:10.3390/ijerph192114202_

Round 1

Reviewer 1 Report

The authors presented an interesting paper, highlighting, through MRI performed at 1, 6, and 15 months, the changes that may occur after physiotherapeutic treatment using ultrasound for three months. The first part of the article presents a clinical case, while the second part contains a theoretical description of possible knee injuries after the US. Elements related to ultrasound treatment, including treatment principles, effects on the bone system, and the role of the US in the therapy of musculoskeletal disorders, are also described.

However, there are a number of issues that need to be checked and corrected. These are some of them:

-          Line 23 and 44 : what”MRI” stands for? Explain the abbreviation.

-          Line 42: remove comma after ”lesions”

-          Line 48: ”therefore” between commas

-          Line 58: ”or”, not ”nor”; comma after ”corticosteroids”

-          Line 59: remove the comma after ”performed”

-          Line 60: ”to instead of ”of”                       

-          Line 68: „minute” insted of ”minutes”;

-          Line 73: consider ”to the” instead of ”for”

-          Line 82: ”in” instead of ”of”; ”reduction” instead of ”decrease”

-          Line 94: ”a” instead of ”an”

-          Line 97: ”non-weight-bearing” instead of ”non-weight bearing”

-          Line 102: consider remove the word ”performed”

-          Line 109: consider change ”in view of” with ”given”; ”treating” instead of ”that we treat”

-          Line 110: remove the comma after ”syndrome”; add ”the” after ”osteonecrosis”; consider remove the word ”lesions”

-          Line 111: ”its” instead of ”their”

-          Line 117: ”syndrome”, not ”syndrom”

-          Line 119: ”sports”, not ”sport”

-          Lines 125-126: ”without” instead of ”in the absence of”

-          Lines 127-128: the explanation of MRI abbreviation already exists

-          Line 132: ”erosion”, not ”erosions”

-          Line 139: ”sports”, not ”sport”

-          Line 140: ”first-line” instead of ”first line”

-          Line 141: ”recommend” instead of ”recommand”

-          Line 143: consider removing ”the” before physiotherapy

-          Lines 152: ”fractures” instead of ”fracture”

-          Lines 156-157: please explain ”.....” from ” due to systemic 156 diseases or treatment side-effects…”

-          Line 159: put ”a” before ”high”

-          Lines 160-161: instead of ” osteonecrosis of the knee” put ”knee osteonecrosis”

-          Line 162: instead of ”at early” put ”in the early”

-          Line 163: remove ”a” before ”conservative”

-          Line 176: put ”hand” after ”other”

-          Lines 180-181: remove ”an” before ”anti-inflammatory”

-          Line 190: instead of ” a number of” put ”several”

-          Line 195: remove ”the” before therapeutic

-          Line 196: remove ”therapeutic” before  ”treatments” ; the words ”effect/effects” are repeating; consider change it

-          Line 198: ”low-intensity” instead of ”low intensity”

-          Line 201: ”non-unions” instead of ”non-union”

-          Line 203: ” remove ”to carry out” from the sentence

-          Line 205: ”a”  instead of ” the” before ” lack”; instead of ” for” put ”of”; ”low-intensity” instead of ”low-intensity”

-          Line 216: put ”.” after ”vs”

-          Line 220: remove comma after ”recommendations”

-          Line 236: ”the” before ”maximum”; ”of” after ”average”

-          Line 238: remove ”to be”

-          Line 245: consider ”the” instead of ”a” before ”beneficial”

-          Line 259: ”eight” for ”8”

-          Line 266: consider removing ”clearly”

-          Line 267: ”aligns” instead of ”is in line”

-          Lines 274-276: consider split the phrase ” Preclinical models have also been studied on this topic: in their study of rabbit femurs, Smith et al. showed that focused US directed at the femur caused immediate and significant thermal damage to the bone, in the form of osteocyte necrosis.”

-          Line 275: ”the” before US

-          Line 276: ”therefore” between commas

-          Line 279: instead of ”periods of time” write  just periods”

-          Line 283: ”a US”, not ”an US”

-          Line 288: ”the current literature” without ”the”

-          Line 291: instead of ”come to the conclusion” write ”concluded”

-          Line 292: ”the” before US

-          Lines 297-299: split the phrase ” In France, there are no formal guidelines concerning the 297 use of US therapy in sports medicine, in myotendinous pathologies, and despite this, this therapy remains widely used by a significant number of physiotherapists.”

-          Line 300: instead of ”as a result of” write ”due to”

-          Line 304: remove ”the” before ”bone”

-          Line 309: change ”in order to” with ”to”

I would also like to add the following:

-US  (ultrasound) – once the abbreviation is explained, it is used and subsequently

-DP FS – explain the abbreviations

-ITB (iliotibial band) – once the abbreviation is explained, it continues with the same abbreviation throughout the entire paper

-Not all references are presented in the form required by the journal

-There are few titles in the last 5 years (within the references)

Author Response

Reviewer Remarks

Authors’ Responses

Reviewer 1.

The authors presented an interesting paper, highlighting, through MRI performed at 1, 6, and 15 months, the changes that may occur after physiotherapeutic treatment using ultrasound for three months. The first part of the article presents a clinical case, while the second part contains a theoretical description of possible knee injuries after the US. Elements related to ultrasound treatment, including treatment principles, effects on the bone system, and the role of the US in the therapy of musculoskeletal disorders, are also described.

However, there are a number of issues that need to be checked and corrected

Thank you for your comments Reviewer 1.

1.      English language corrections

Thank you for your corrections. We have implemented all of them in the manuscript.

2.      Abreviation modifications

Thank you for your corrections.

3.      Not all references are presented in the form required by the journal

I hope we have made the necessary changes considering reference presentation.

Reviewer 2 Report

This study reports a case with bony lesions potentially induced from ultrasound therapy and reviews available literature on therapeutic effect of US and its association with bony lesions. While the case report is well-written overall, please see below for my suggestions and comments.

IN GENERAL

Can you provide evidence behind US treamtent on IT band syndrome?

INTRODUCTION

I would rearrange paragraph 2 (Line 41-44) and paragraph 3 (Line 45-49) so that paragraph 3 comes first. And perhaps merge the current paragraph with the last paragraph in a way that a brief overview of the case is provided as well as the aim of this case report.

CASE REPORT

Line 55-56 what type of exercise?

Line 66-67: different providers perform different management but can you provide reference to support the rationale for this treatment? Especially, why 20 US sessions were planned 

Line 98. What was done after US session was halted? Can you provide more specifics on rehabilitation process between the beginning of treatment and 15 months period? Any activity restriction during this time? Were there any other potential causes of this MRI findings? Why do you think this finding was reversed?

DISCUSSION

Line 109. Can you clarify what you mean by "we treat the main aspects of US therapy."? 

Line 120. There are several proposed etiology/mechanisms of IT band syndrome. Excessive friction is not always the case based on the anatomical study. Pain may arise from pain-sensitive fat tissue beneath the IT band. Please discuss several etiology/mechanism here.

Line 157. Rather than ..., add etc.

Line 166-167. Again, other than stopping US, what else was done?

CONCLUSION

Line 309-310: I disagree with this conclusion because as the authors mentioned there is already very low evidence of therapeutic effect of ultrasound and trying to find an optimal dose and frequency would rather cause adverse bone health than anything else.  I would conclude with warning that despite wideuse of US therapy session, there is limited evidence and it can rather induce bony lesions. 

Author Response

Reviewer 2.

This study reports a case with bony lesions potentially induced from ultrasound therapy and reviews available literature on therapeutic effect of US and its association with bony lesions. While the case report is well-written overall, please see below for my suggestions and comments.

Thank you for your comments Reviewer 2.

1.      I would rearrange paragraph 2 (Line 41-44) and paragraph 3 (Line 45-49) so that paragraph 3 comes first. And perhaps merge the current paragraph with the last paragraph in a way that a brief overview of the case is provided as well as the aim of this case report.

Thank you for your remark.

We have modified the text as you request.

“The widespread use of US therapy, with no known dose-response relationship, is an issue. US therapy users can only guess what dose might be useful for a patient and how to modify it. The level of clinical benefit to the patient from US physiotherapy treatments therefore remains uncertain, and more clinical research is needed to justify the dosages currently used in treatment [4,5].

There are few articles related to the side effects, more precisely bone abnormalities or “osteonecrotic-like” lesions of US used for therapeutic purposes. Here we report an example of a lesion that did not lead to clinical consequences but instead to the appearance of a noticeable unusual abnormality on MRI. Our case report and short review of the literature thus focus on these radiological abnormalities following US therapy. “

2.      Line 55-56 what type of exercise?

Thank you for your remark, we have completed this part:

“She had been complaining for two months of pain on exertion in the context of tendinopathy of the ITB syndrome type secondary to practicing indoor exercise (musculation, cross-fit,…).”

3.      Line 66-67: different providers perform different management but can you provide reference to support the rationale for this treatment? Especially, why 20 US sessions were planned 

The sessions were carried out by the physiotherapist without following any particular recommendation.

There is therefore no justification on his part for carrying out 20 sessions.

4.      Line 98. What was done after US session was halted? Can you provide more specifics on rehabilitation process between the beginning of treatment and 15 months period? Any activity restriction during this time? Were there any other potential causes of this MRI findings? Why do you think this finding was reversed?

Thank you for your remark.

“Due to chronology, these lesions were utterly compatible with bone remodeling connected with US. In fact, these are lesions occurring in a non-weight-bearing area next to the treated area. No other support in rehabilitation was carried out after the ultrasounds and no restriction after the end of the ultrasound sessions.”

To our knowledge, no other cause can explain these lesions in this patient.

5.      Line 109. Can you clarify what you mean by "we treat the main aspects of US therapy."? 

Thank you for your remark.

We wanted to introduce our discussion in order to talk about US therapy in more detail. We have modified this sentence as requested:

“This specific case leads to some questions about US therapy”

6.      Line 120. There are several proposed etiology/mechanisms of IT band syndrome. Excessive friction is not always the case based on the anatomical study. Pain may arise from pain-sensitive fat tissue beneath the IT band. Please discuss several etiology/mechanism here.

Thank you for your remark, we have completed this part

“It is usually seen in people with intense physical activity, such as athletes. It seems to be the most common running injury of the lateral knee region, with an incidence of 1.6-12% [8,9]. The kinetics and kinematics of the hip, knee and ankle also seem to play an important role in ITB syndrome and neuromuscular coordination emerged as a likely reason for the kinematic defects."

7.      Line 309-310: I disagree with this conclusion because as the authors mentioned there is already very low evidence of therapeutic effect of ultrasound and trying to find an optimal dose and frequency would rather cause adverse bone health than anything else.  I would conclude with warning that despite wide use of US therapy session, there is limited evidence and it can rather induce bony lesions. 

Thank you for your remark.

We have modified the text as you request.

“In conclusion, this case suggests US therapy may be associated with localized "osteonecrosis-like" bony lesions, as seen on MRI. Although these lesions were not associated with any clinical symptoms, and disappeared over time once the US treatment had stopped. Thus, despite wide use of US therapy, there is limited evidence of its effectiveness and it may instead induce bone damage.”

Round 2

Reviewer 1 Report

I have carefully read the new version of the paper and the changes made and I believe that the article can be published in the current form.

Author Response

Reviewer 1.

I have carefully read the new version of the paper and the changes made and I believe that the article can be published in the current form.

Thank you for your comments Reviewer 1.

Reviewer 2 Report

The authors did a decent job in improving the manuscript.

I strongly recommend reviewing the article (especially anatomical consideration part) https://www.sciencedirect.com/science/article/abs/pii/S1934148211000268

and add on line Line 119-123 that IT band syndrome does not always happen as a result of friction. 

Author Response

Reviewer 2.

The authors did a decent job in improving the manuscript.

Thank you for your comments Reviewer 2.

I strongly recommend reviewing the article (especially anatomical consideration part) https://www.sciencedirect.com/science/article/abs/pii/S1934148211000268

The first insertion of the iliotibial band is in the distal femur at the upper edge of the lateral epicondyle. The second attachment of the iliotibial band is the insertion into Gerdy's tubercle of the tibia and serves as a ligament in its structure and function. The iliotibial band has many other distal attachments (the biceps femoris, vastus lateralis, lateral patellar retinaculum, patella, and patellar tendon). The site of injury is often associated with insertion at the lateral epicondyle, but interrelated with the forces created by the various attachments above and below the lateral epicondyle.

add on line Line 119-123 that IT band syndrome does not always happen as a result of friction. 

Thus, several intrinsic and extrinsic contributing factors to ITBS have been described. Reduced hip muscle performance and abnormal hip and knee mechanics during functional tasks may be major contributors to ITBS
